# Satellite Lidar Measurements as a Critical New Global Ocean Climate Record

Michael J. Behrenfeld [1,*], Laura Lorenzoni [2], Yongxiang Hu [3], Kelsey M. Bisson [1,2], Chris A. Hostetler [3], Paolo Di Girolamo [4], Davide Dionisi [5], Francesco Longo [6] and Simona Zoffoli [6]

1 Department of Botany and Plant Pathology, Oregon State University, Corvallis, OR 97331, USA; bissonk@oregonstate.edu
2 Ocean Biology and Biogeochemistry Program, Earth Science Division, Science Mission Directorate, NASA Headquarters, Mail Suite 3Y35, 300 E St., SW, Washington, DC 20546-0001, USA; laura.lorenzoni@nasa.gov
3 NASA Langley Research Center, MS 475, Hampton, VA 23681-2199, USA; yongxiang.hu-1@nasa.gov (Y.H.); chris.a.hostetler@nasa.gov (C.A.H.)
4 School of Engineering, University of Basilicata, 85100 Potenza, Italy; paolo.digirolamo@unibas.it
5 Institute of Marine Sciences, National Research Council (ISMAR-CNR), 00133 Rome, Italy; davide.dionisi@artov.ismar.cnr.it
6 Italian Space Agency (ASI), 00133 Rome, Italy; francesco.longo@asi.it (F.L.); simona.zoffoli@asi.it (S.Z.)
* Correspondence: michael.behrenfeld@oregonstate.edu; Tel.: +1-541-737-5289

**Abstract:** The year 2023 marked the tenth anniversary of the first published description of global ocean plankton stocks based on measurements from a satellite lidar. Diverse studies have since been conducted to further refine and validate the lidar retrievals and use them to discover new characteristics of plankton seasonal dynamics and marine animal migrations, as well as evaluate geophysical products from traditional passive ocean color sensors. Surprisingly, all of these developments have been achieved with lidar instruments not designed for ocean applications. Over this same decade, we have witnessed unprecedented changes in ocean ecosystems at unexpected rates and driven by a multitude of environmental stressors, with a dominant factor being climate warming. Understanding, predicting, and responding to these ecosystem changes requires a global ocean observing network linking satellite, in situ, and modeling approaches. Inspired by recent successes, we promote here the creation of a lidar global ocean climate record as a key element in this envisioned advanced observing system. Contributing to this record, we announce the development of a new satellite lidar mission with ocean-observing capabilities and then discuss additional technological advances that can be envisioned for subsequent missions. Finally, we discuss how a potential near-term gap in global ocean lidar data might, at least partially, be filled using on-orbit or soon-to-be-launched lidars designed for other disciplinary purposes, and we identify upcoming needs for in situ support systems and science community development.

**Keywords:** satellite lidar; ocean ecosystems; ocean monitoring

## 1. Introduction

Satellite remote sensing of global ocean ecosystems began with the Coastal Zone Color Scanner (CZCS; 1978–1986) and, following a decade-long gap in the ocean color climate record, has since been continued through a series of overlapping measurements from the Ocean Color and Temperature Sensor (OCTS; 1996–1997), Sea-viewing Wide Field-of-view Sensor (SeaWiFS; 1997–2010), Moderate Resolution Imaging Spectroradiometers (MODIS-Aqua; 2002-present), Medium Resolution Imaging Spectrometer (MERIS; 2002–2012), and other subsequent satellite sensors [1,2]. Insights into ocean ecosystem functioning and responses to environmental change enabled through these measurements have been so profound that sustaining this record is now viewed as essential for satisfying operational, research, and societal needs. Indeed, the Global Ocean Observing System (https://www.

goosocean.org/, accessed on 15 August 2023) and Global Climate Observing System (https://gcos.wmo.int/en/home, accessed on 15 August 2023) programs identify ocean color as an essential ocean and climate variable supporting the monitoring of ocean health, fisheries, and other aspects of marine ecosystems and climate. Provision of these data and ensuring an unbroken record into the foreseeable future has been and will continue to be an international undertaking.

The ocean color technique relies upon the influence of dissolved and particulate materials on the intensity and spectral properties of backscattered sunlight emanating from surface waters and, thus, is referred to as a 'passive remote sensing' approach. A completely different and complementary approach is provided through the 'active' light detection and ranging (lidar) technique. A lidar is designed to measure scattering signals from a laser pulse rather than sunlight. These return signals are recorded at a very high sampling rate (e.g., $10^7$–$10^8$ samples per second) and this rapid time-of-flight recording allows retrieval of range-resolved properties (e.g., height in the atmosphere or depth in water) [3]. In situ and airborne lidar systems have a long history of aquatic applications [3,4] and recently, ocean property retrievals have been demonstrated using satellite lidar [5–20]. The motivation for the current manuscript is not to overview results from these earlier studies but rather to outline the new vision inspired by these successes regarding satellite lidar observations as a key expansion to the global ocean climate data record. If this vision is realized, its value to Earth system science goes well beyond aquatic systems and includes atmosphere, terrestrial, and cryosphere sciences, although these applications are not discussed in any detail herein.

## 2. Lidar Advantage

Satellite ocean color measurements have revolutionized our understanding of aquatic ecosystems and evidenced their complex interactions with other facets of the Earth system. They have provided the scientific community with the ability to assess phytoplankton abundance, composition, and health through multispectral and hyperspectral measurements and to distinguish various water constituents in dynamic areas such as coastal zones. Ocean color spectral inversion algorithms, such as the Garver–Siegel–Maritorena (GSM) algorithm [21–23], Quasi-Analytical Algorithm (QAA) [24], and Generalized Inherent Optical Properties (GIOP) algorithm [25], have provided the oceanographic community with global fields of attenuation ($K_d$) and backscattering ($b_{bp}$) data for decades. So, where do satellite ocean lidar retrievals come in and what is their significance for communities that work with and rely on aquatic ecosystems? The answer to these questions is manifold.

The satellite ocean lidar era began with the Cloud-Aerosol Lidar with Orthogonal Polarization (CALIOP) sensor [26], an instrument never intended for measuring ocean properties. CALIOP was one element of the A-train Earth Observing Sensor suite (https://atrain.nasa.gov/, accessed on 1 August 2023), which was jointly developed by the National Aeronautics and Space Administration (NASA) and the Centre National d'Etudes Spatiales (CNES) and designed to characterize height-resolved aerosol and cloud properties of the atmosphere. Nevertheless, the neodymium yttrium aluminum garnet (Nd:YAG) lasers employed by CALIOP provided a linear polarized emission at 532 nm that effectively penetrated into the ocean and yielded depth-integrated information on plankton properties of the upper surface layer (predominantly < 20 m depth) through the ratio of the measured cross- and co-polarized signal relative to the laser emission [5,13,27]. The ocean optical property retrieved most directly by CALIOP is termed 'attenuated-backscatter' because its strength is determined both by the efficiency with which the laser pulse is backscattered toward the satellite telescope and the two-way attenuation of photons through the intervening atmosphere and water. While the simple elastic backscatter technology of CALIOP did not allow the separation of $K_d$ and $b_{bp}$ directly, various approaches have been implemented to retrieve these coefficients from the measured signal, as reviewed in [27].

Space-based retrievals of $b_{bp}$ data have led to new perspectives on aquatic ecosystems. Since it is an active measurement and thus not dependent on sunlight, lidar can

observe the ocean both day and night. This means that polar ocean ecosystems, which are currently among the most impacted by climate warming [28–30], can be monitored throughout the annual cycle (Figure 1A), whereas ocean color data for these regions may be non-existent for months on end (Figure 1B). This lidar advantage can improve understanding of key ecological events occurring from late autumn to early spring, such as initiation of the annual phytoplankton bloom, e.g., [6]. The range-resolved nature of lidar measurements also means that detailed optical properties of the overlying atmosphere are uniquely resolved, allowing accurate isolation of the ocean signal even through significant aerosol loads and thin clouds [3,5,11,27,31]. By contrast, atmosphere and ocean signals are convoluted in passive ocean color measurements, which compromises ocean property retrievals in the presence of aerosols and clouds due to uncertainties in applied 'atmospheric correction' schemes [32–35]. Furthermore, the narrow viewing geometry of satellite lidar measurements allows accurate ocean retrievals through small cloud gaps that would otherwise prevent valid ocean color retrievals that require cloud-free conditions and are subject to errors resulting from 'adjacency effects' (i.e., signal contamination from nearby clouds) [36,37]. This lidar advantage is thus particularly important for ocean observing in persistently cloudy areas, such as polar regions, upwelling zones, and the intertropical convergence zone [38,39].

Perhaps one of the most powerful attributes of space-based lidars is their ability to observe global ocean ecosystems in three dimensions [3]. Passive ocean color remote sensing can only detect depth-integrated signatures of plankton communities very near the ocean surface, thus missing significant ecosystem structuring at greater depths. While in situ measurements of depth-dependent plankton distributions have been achieved with ship-based and autonomous platforms, these in situ data remain sparse and leave large uncertainties regarding contemporary ocean functioning and prediction of future change. Recent characterization of global patterns in diel vertical animal migrations [7] (Figure 1C) using CALIOP measurements has expanded ocean observing capabilities and opened tantalizing possibilities for further depth-resolved assessments of these vital animal populations and all that depend on them. Importantly, this diel migration plays a major role in mid- to long-term ocean carbon sequestration [40–42] and is a fundamental element of the oceanic food web [43,44]. CALIOP has fortuitously provided a baseline record of global diel migrating animal populations [7] against which future lidar measurements can be compared to evaluate changes to these communities in response to ocean warming and targeted mesopelagic fishing activities.

Satellite lidar measurements also provide independent global observations of ocean properties that can be compared to passive ocean color products. The traditional approach to evaluating geophysical products from ocean color algorithms has been through comparison with ship measurements [45], but these in situ data are, again, relatively rare and biased in spatial coverage (Figure 1D) [34]. In contrast, satellite lidar measurements provide a globally representative sampling of the ocean that, within even a single month, is significantly greater than the coverage provided by decades of ship measurements (Figure 1E). Bisson et al. [46,47] demonstrated the value of independent satellite lidar data by first showing the improved accuracy of $b_{bp}$ retrievals from CALIOP compared to MODIS-Aqua (a finding later echoed by Sun et al. [17]) and then used the lidar data to reveal previously unrecognized seasonal biases in ocean color $b_{bp}$ products that span multiple ocean color missions (Figure 1F). Behrenfeld et al. [5] compared CALIOP and MODIS-Aqua retrievals of $b_{bp}$, phytoplankton carbon, and total particulate organic carbon products to highlight regional similarities and biases among ocean color-inversion algorithms. More recently, Bisson et al. [48] demonstrated how lidar $b_{bp}$ data can be used to 'seed' an ocean color-inversion algorithm to improve assessments of other properties retrieved from ocean color data, such as phytoplankton absorption coefficients ($a_{ph}$). The provision of global and independent ocean properties from lidar is an advantage that cannot be overstated, as it allows separate and robust validation of critically important products [20,46,47] that are

utilized far beyond research, informs directions for future improvement, and ultimately advances Earth system understanding.

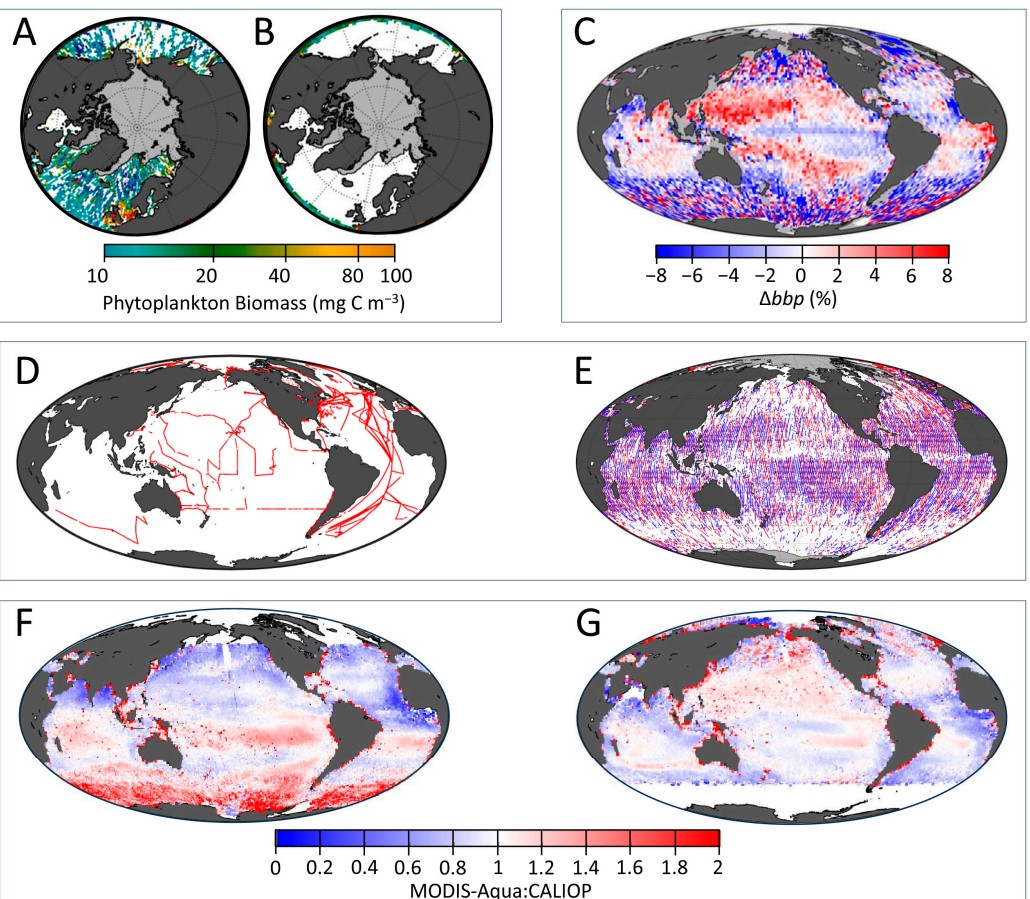

**Figure 1.** Lidar advantage. (**A**,**B**) Comparison of spatial coverage in north polar region phytoplankton biomass for December 2010 using (**A**) CALIOP lidar measurements and (**B**) MODIS-Aqua ocean color measurements from [6]. (**C**) Annual average signal (2008–2017) of diel vertically migrating animals detected using CALIOP lidar measurements as a day–night difference in $b_{bp}$ from [7]. (**D**,**E**) Comparison of global coverage in independent $b_{bp}$ measurements available for comparison with ocean color products. (**D**) All $b_{bp}$ data available from the NASA SeaBASS data archive (including observations from ship overboard casts and flow-through measurements and mooring data) and collected over ~17.5 years (2004–2021; total data locations > 1,300,000). (**E**) Valid CALIOP lidar ocean $b_{bp}$ retrievals for a single representative month (April 2010). Red symbols indicate daytime retrievals (total data locations > 2,300,000) and blue symbols indicate night-time retrievals (total data locations > 1,800,000). Note the regional bias of SeaBASS data compared to the globally representative coverage of CALIOP data. (**F**,**G**) The 2006–2017 average seasonal bias in ocean color-based $b_{bp}$ compared to CALIOP $b_{bp}$, for example months of (**F**) January and (**G**) July [47]. Gray pixels in (**A**,**B**,**E**) indicate ice cover. Gray pixels in (**C**) correspond to shallow water bathymetry mask.

The unique perspective that lidar measurements bring to aquatic sciences and their complementarity to ocean color measurements are instrumental for reducing uncertainties in ocean plankton stocks and elemental cycles and for documenting how these properties are changing in response to stressors. While the lidar technique clearly provides many advantages for improved understanding and monitoring of the global ocean, it also has limitations (e.g., near-nadir viewing only and limited spectral resolution). As our global ocean ecosystems are confronted with an increasing number of compounding stressors, including warming temperatures and acidification [49,50], strengthening water column stratification and associated shifts in nutrient stress [51–53], expanding fisheries exploitations [54–59],

increasingly frequent regional-scale extreme events (e.g., marine heat waves) [60–62], thinning and retreat of seasonal sea ice [63], and proliferation of plastics pollution [64,65], a sustained and complementary global observing system is required to quantify, predict, and mitigate associated impacts. Such an observing system must be inclusive of satellite lidar, passive ocean color, and polarimetry measurements [2,4,66], modeling, and in situ observations. The era of single-platform measurements is over. A new era must begin where the seamless integration of multiple ocean-observing technologies provides multidimensional observations of aquatic ecosystems.

## 3. On the Horizon—Vision for a Lidar Era in Oceanography

Future satellite lidar missions have the potential to retrieve a variety of important new ocean properties that have already been demonstrated using airborne lidar [3]. For example, a UV-VIS satellite lidar can quantify absorption by colored dissolved organic material (CDOM), provide information on the size distribution of suspended particles in the ocean, and enhance retrievals over a broader range of water types [67,68]. Faster detector sampling speeds can enable water column profiling of plankton properties (Figure 1E) and, in conjunction with ocean color data, permit four-dimensional reconstructions of ocean ecosystems. Expansion of detector spectral bands will significantly improve the accuracy of retrieved geophysical properties (e.g., dissection of attenuated backscatter into $b_{bp}$ and $K_d$ (see below)) [3,34], provide new information on phytoplankton 'health' and community composition [69–71] through lidar measurements of night-time stimulated chlorophyll fluorescence, and potentially furnish an opportunity to map suspended plastics pollution [72,73]. Future depth-resolved global lidar data will also yield an opportunity to significantly improve estimates of ocean net primary production (NPP) through the provision of vertically resolved plankton distributions below the ocean color detection depth [8,74], allow more accurate separation of absorption by phytoplankton pigments and colored dissolved organic matter [3,75], and enable independent assessments of plankton stocks for refining ocean color algorithms [48]. A future constellation of low-cost lidar would be able to observe diel cycles in $b_{bp}$ that help constrain estimates of phytoplankton division rates [76–79], among other applications.

Looking broader, repeated global lidar observations of depth-resolved optical properties can aid in assessing upper ocean active mixing layers [80], understanding plankton distributions in the context of ocean physics, and quantifying materials exchange (for example, particulate carbon) between depth layers and export to the deep sea (particularly in conjunction with in situ assets, such as a global fleet of profiling autonomous floats). Finally, satellite lidar observations have a variety of applications for unique habitats and conditions, including near-shore bathymetry [81] and submerged vegetation mapping [82], water quality assessments, characterization of plankton properties both below sea ice and along the ice–water interface [11,83], and surface oil sensing [84]. Through interdisciplinary collaboration, lidar profiles of atmospheric properties can also contribute critical information for improving ocean color atmospheric correction approaches.

These landmark advances in the observation of below-surface ecosystems have heightened awareness of the susceptibility of ocean ecosystems to change and simultaneously highlighted the limitations of current observations essential for achieving robust predictions. There is an urgent need to not just continue past sensor capabilities but to take the next step in aquatic lidar observations from space. Addressing this need soon is critical to minimize observational gaps in the global lidar climate record, as such gaps can have significant impacts on assessing temporal trends and assigning time of emergence to these trends. Recognizing this urgency, the Italian Space Agency (ASI) and NASA are considering a partnership on a new mission with a launch readiness date for the end of the decade. The primary payload for this mission would be the Cloud Aerosol Lidar for Global Observations of the Ocean–Land–Atmosphere (CALIGOLA) instrument (Figure 2). With its rapid development and launch goals, the CALIGOLA design would be based on high technology readiness level (TRL) components.

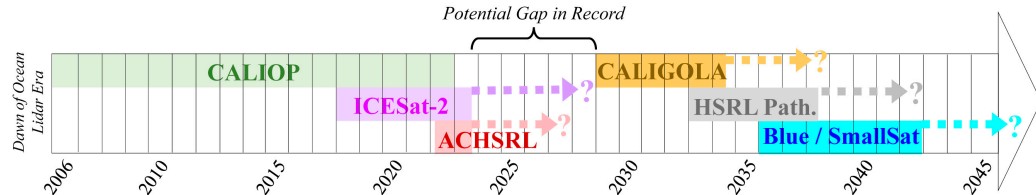

**Figure 2.** Creating a sustained global climate data record of satellite lidar ocean observations. The record began with CALIOP in 2006 and is currently being continued in part by the ICESat-2 and ACHSRL missions. The upcoming CALIGOLA mission could provide enhanced ocean observations, but final architecture of this mission is yet to be determined. The NASA HSRL Pathfinder is being designed specifically for enhanced ocean-observing capabilities, which could be further augmented using blue laser technology potentially aboard a SmallSat platform.

The CALIGOLA instrument is currently in its first phase of implementation, so its detailed architecture is not yet finalized, but notional plans have been developed. Building upon Aeolus wind lidar and the Earth Cloud Aerosol and Radiation Explorer (EarthCARE) atmospheric lidar heritage, the CALIGOLA laser system will include three linear polarized emission wavelengths: 354.71 nm (~150 mJ), 532.05 nm (~45 mJ), and 1064.10 nm (~155 mJ) (abbreviated hereafter as 355, 532, and 1064 nm), with emissions of 50 to 100 pulses per second (i.e., 75 to 150 m horizontal distance between pulses at the Earth's surface). The 355 nm and 532 nm emissions are the critical ocean-penetrating wavelengths.

Laser pulse return signals collected by CALIGOLA's telescope will be partitioned across a set of detector channels. Three of these channels will measure co-polarized signals at the emission wavelengths, while three additional channels will measure cross-polarized signals at these same wavelengths. While CALIOP's coarse sampling approach limited the retrieval of $b_{bp}$ to a depth-integrated value weighted toward the surface similar to ocean color, CALIGOLA's off-nadir viewing geometry and advanced detection approach will allow depth-resolved measurements with both 355 and 532 nm co- and cross-polarizations, providing richer observations of ocean optical properties. CALIGOLA retrieval depths will be enhanced by the far greater laser power at 355 nm compared to the power of either CALIOP or CALIGOLA emissions at 532 nm. Simultaneous measurements of the cross-polarized signals at the emission wavelengths allow direct comparison between CALIGOLA and CALIOP records, characterization of particle types and plankton community composition [85], and potentially the retrieval of beam attenuation coefficients for the ocean (a valuable property for quantifying plankton standing stocks) [86].

Beyond the six emission-aligned detector channels described above, a variety of additional detector channels are being considered for CALIGOLA. One of these candidate channels is focused on measuring Stokes (404–409 nm) vibrational water Raman scattering signals. This Raman channel would be instrumental for partitioning attenuated backscatter measured by the 355 nm and 532 nm channels into its two fundamental terms. Specifically, the Raman channel signal is sensitive to molecular water density (which is known) and insensitive to the hydrosol concentration; hence, the attenuation of the lidar signal provides a measure of the attenuation coefficients ($K_{d,\lambda}$) at 404–409 nm. This independent $K_{d,\lambda}$ retrieval can then be spectrally extended to $K_d$ at 355 nm and 532 nm and used to directly separate attenuation and backscattering elements of the co-polarized and cross-polarized signals measured at these latter wavelengths. The combination of the backscatter and Raman channels also enables measurements of snow depth, snow density, and snow water equivalent [86].

Also being considered for CALIGOLA are two additional detector channels with ocean applications. The first of these is a 680 nm channel for measuring chlorophyll fluorescence. Pigments in phytoplankton effectively absorb excitation energy at 355 nm and 532 nm and pass this energy to chlorophyll molecules [87]. Most of this energy is used for photosynthesis or dissipated as heat, but some of it is remitted as fluorescence. As noted above, the strength of this fluorescence signal is dependent on upper ocean growth conditions (e.g., presence or absence of iron stress [70]), and thus its measurement

with a satellite lidar (particularly at night) provides a means to assess the 'health' of phytoplankton communities. This chlorophyll fluorescence channel would also provide critical information on terrestrial plant health and photosynthesis [88–90]. The other ocean-relevant CALIGOLA detector channel would be centered at ~455 nm to allow assessments of CDOM fluorescence, surface oils, and plastics pollution. Finally, two rotational Raman channels are being considered for the CALIGOLA detector system that, in conjunction with the suite of detectors described above, would enable major advances toward atmospheric science objectives (Box 1).

**Box 1.** CALIGOLA atmospheric science advances.

> CALIGOLA's three co-polarized and three cross-polarized emission-aligned detector channels with sampling rates equivalent to <1 m vertical resolution will extend and enhance upon heritage atmospheric products provided by CALIOP and the Atmospheric Lidar (ATLID) instrument of the EarthCARE mission. In addition, CALIGOLA will nominally include two pure-rotational Raman channels for measuring scattering from atmospheric nitrogen and oxygen molecules stimulated by the UV laser pulses at 354.71 nm. Each of these channels will include temperature-insensitive lines, but with one aligned on the Stokes (355.7 nm) and the other anti-Stokes (353.95 nm) branches. The possibility of including temperature-sensitive lines is also under consideration. The rotational Raman channels are of paramount importance for providing a reference signal to accurately infer atmospheric particle (aerosol/clouds) backscattering coefficient profiles at 354.71, 532.05, and 1064.1 nm (i.e., $\beta_{355}(z)$, $\beta_{532}(z)$, and $\beta_{1064}(z)$, respectively) from the corresponding elastic signals [91,92]. The rotational Raman signals can further be used to determine atmospheric particle extinction coefficient profiles at 354.71 nm (i.e., $\alpha_{355}(z)$) when used in conjunction with atmospheric thermodynamic profiles from radiosondes or models [93]. Finally, the combination of rotational Raman and co-polarized and cross-polarized elastic signals allows the assessment of atmospheric particle (aerosol/clouds) depolarization ratios at 354.71, 532.05, and 1064.1 nm (i.e., $\delta_{355}(z)$, $\delta_{532}(z)$ and $\delta_{1064}(z)$, respectively).
>
> In addition to the atmosphere-focused rotational Raman channels, the vibrational Raman channels described in the main text can capture signals from water vapor in the atmosphere, liquid water in the ocean, and water in snow and ice. Raman scattering wavelengths differ slightly between these water-aggregation phases and, thus, the performance of CALIGOLA in measuring these signals will depend on the final detector design. Finally, the ~455 nm channel described in the main text for CDOM, oil, and plastics detection can also be used to determine the atmospheric particle (aerosol) fluorescence coefficient (i.e., $\beta_{FL\_AER}(z)$), allowing the characterization of atmospheric aerosols with organic components (e.g., biological particles, biomass fuels, sulfates, and dust).

In its current conception, all ocean-relevant channels would employ high TRL detectors with high sampling rates equivalent to <1 m vertical resolution. The vertical resolution of retrieved ocean properties will, nevertheless, differ between channels in accordance with respective signal strengths. For example, the weaker fluorescence and Raman signals may be detectable only within the first optical depth of the water column and require integration throughout this layer to enhance the signal-to-noise ratio (SNR). Even so, these surface layer properties are expected to be representative of the entire active surface mixing layer. Much stronger return signals are expected for the other ocean channels described above, thus requiring minimal vertical averaging to achieve sufficient SNR and allowing property retrievals at meter-scale vertical resolution. For example, with appropriate depth-dependent horizontal averaging, it may be anticipated that valid retrievals are achieved to ~2 optical depths for the cross-polarized 532 nm channel, ~3 optical depths for the 532 nm co-polarized and 355 cross-polarized channels, and potentially ~4 optical depths for the 355 nm co-polarized channel.

At the observatory level, CALIGOLA is currently expected to provide similar global ocean sampling coverage to CALIOP [27]. Unlike CALIOP, the CALIGOLA sensor is not expected to fly in formation with any specific ocean color sensor (note: CALIOP and MODIS-Aqua were both part of the A-train) but instead would overlap measurements with multiple ocean color missions (see https://ioccg.org/resources/missions-instruments/scheduled-ocean-colour-sensors/, accessed on 1 August 2023). This arrangement would be beneficial for intercomparisons of lidar and ocean color products as it ensures that

lidar data are available for matchups with ocean color measurements over a wide range of viewing angles, thereby maximizing opportunities to identify scan-angle discrepancies between sensors.

If executed as described above, CALIGOLA will extend the climate data record of CALIOP-equivalent ocean measurements and initiate new climate data records addressing many of the expanded science objectives identified above in Section 2 (Table 1). However, while CALIGOLA lasers may have the potential to operate for over a decade, the mission currently has a 3-year design lifetime and carries a planned 2-year extension (note: CALIOP also had a 3-year design lifetime yet operated for 18 years). Given the long time typically required to mature a new satellite mission from pre-Phase A to launch, it is therefore critical that the international science community begins planning now for follow-on missions to ensure an unbroken future continuation of the satellite lidar ocean climate data record. While the rapid development of CALIGOLA has demanded planning based on high TRL components, future lidar missions may employ more advanced technologies that improve retrieval accuracies and expand science applications.

One example of an advanced future mission is the High-Spectral-Resolution Lidar (HSRL) Pathfinder being developed at the NASA Langley Research Center. The HSRL Pathfinder is designed as a cost-effective approach to achieving cross-cutting aerosol, cloud, and ocean observations recommended in the 2017 Decadal Survey for Earth Science and Applications from Space (https://www.nationalacademies.org/our-work/decadal-survey-for-earth-science-and-applications-from-space, accessed on 13 October 2021). The lidar would operate at the same polarization sensitivity and 532 nm and 1064 nm emission wavelengths as CALIOP, but would incorporate two key advances: (1) high vertical resolution sampling (~1 m) similar to CALIGOLA and (2) inclusion of high-spectral-resolution lidar capability [94–96] at 532 nm, which enables direct and independent retrieval of $b_{bp}$ and $K_d$ profiles [3] (Table 1). In addition to the fundamental importance of these parameters, as already noted, retrieving these profiles will significantly improve NPP estimates and allow depth-resolved characterization of plankton community composition, as demonstrated using similarly capable airborne HSRL lidars [74,85]. HSRL capability also significantly improves the characterization of aerosols and clouds, which addresses atmospheric science objectives and makes the data more valuable for advancing atmospheric correction schemes in ocean color algorithms. While the HSRL Pathfinder concept did not include a chlorophyll fluorescence channel, that channel could be added with little impact on overall instrument cost. Significant investment has already been made in HSRL Pathfinder technology development [97], making this type of instrument viable for a near-term satellite mission (Figure 2). Mass and volume reductions in the design enable its co-deployment with other instruments (e.g., a polarimeter) on a relatively small and cost-effective spacecraft bus, further reducing mission costs by allowing shared launches with other satellites [97].

Another major advance in open-ocean profiling will be made possible by the development of lidar transmitters emitting in the blue wavelength region (e.g., 470–490 nm) (Figure 2, Table 1). Attenuation of the signal from water absorption is much lower, and detector sensitivities are higher, at blue wavelengths than at the 532 nm and 355 nm traditional workhorse laser wavelengths used for space-borne atmospheric profiling and ice-sheet altimetry. HSRLs with blue laser transmitters will enable profiling to significantly greater depths for retrievals of $b_{bp}$ and $K_d$, while still allowing a reduction in lidar mission costs through reductions in power, mass, and size. These reductions enable lower-cost deployment on SmallSats in shared-launch scenarios with other satellites and/or synergistic ocean sensors on the same satellite. Moreover, blue wavelengths have the advantage over 532 nm and 355 nm of being more effective at stimulating chlorophyll fluorescence, and, again, adding a receiver channel for measuring this fluorescence signal has a minimal impact on mission cost. A blue laser operating at the 486 nm Fraunhofer line would have the added benefit of a >3-fold reduction in background noise from scattered sunlight relative to 532 nm, offering further daytime precision and depth penetration advantages, especially for scenes with overlying tenuous clouds for which scattered sunlight can be significant.

**Table 1.** Past, present, and future satellite lidar instruments, measurement capabilities, enabled science (blue text in right-most column), and strengths/challenges (black text in right-most column) contributing to a publicly available lidar global ocean climate data record.

| Instrument | Ocean Relevant Technical Characteristic | Enabled Science Strengths/Challenges |
|---|---|---|
| CALIOP | 532 nm cross-polarized channel<br><br>532 nm co-polarized channel<br><br>Detectors: low sensitivity/~20 m vertical sampling<br><br>Nadir viewing<br><br>16-day orbit repeat cycle | • First global ocean plankton observations with satellite lidar, unprecedented observations of polar systems, detection of marine diel migrating animals in surface ocean, and intercomparisons with ocean color products.<br>• Nadir viewing and detector technology prevented ocean retrievals from 532 nm co-polarized channel.<br>• Day and night ocean measurements of attenuated backscatter from 532 nm cross-polarized channel.<br>• Robust ocean retrievals for first ~20 m bin below ocean surface (no vertical profiling).<br>• Enhanced product uncertainties due to indirect separation of attenuation and backscattering. |
| ATLAS | 532 nm emission channel<br><br>Detectors: high-sensitivity single photon counting/cm-scale vertical sampling<br><br>Nadir viewing<br><br>91-day orbit repeat cycle | • Night-time ocean plankton observations comparable to CALIOP but vertically resolved within the upper 20 m of the surface ocean, demonstrated regionally but with some global potential.<br>• Daytime ocean retrievals compromised by sunlight contamination, thus prohibiting assessments of migrating marine animals.<br>• Enhanced product uncertainties due to indirect separation of attenuation and backscattering. |
| CALIGOLA | 532 nm cross-polarized channel<br><br>532 nm co-polarized channel<br><br>355 nm cross-polarized channel<br><br>355 nm co-polarized channel<br><br>404 nm vibrational water Raman<br><br>645 nm vibrational water Raman<br><br>680 nm chlorophyll fluorescence<br><br>455 nm fluorescence channel<br><br>Detectors: high sensitivity/<1 m vertical sampling<br><br>Off nadir tilt<br><br>16-day orbit repeat cycle | • 4-dimensional reconstruction of plankton communities.<br>• Reduced uncertainty in ocean primary production assessments.<br>• Vertically resolved diel marine animal migrations.<br>• Phytoplankton health, CDOM, and plastics pollution of the active surface mixing layer, plus surface oil contamination.<br>• Plankton community composition, size distribution, particle type, and beam attenuation.<br>• Spectral absorption and attenuation.<br>• Ocean physical–biological dependencies.<br>• Improved assessments of passive ocean color data products.<br>• Advanced atmospheric correction schemes for passive ocean color data.<br>• Improved characterization of polar, upwelling, and intertropical convergence zones.<br>• 532 nm cross-polarized channel provides surface ocean properties directly comparable to CALIOP record.<br>• Enhanced power of 355 nm co-polarized channel extends ocean property retrievals deep into water column (~4 optical depths).<br>• Reduced product uncertainties through Raman-based direct separation of attenuation and backscattering. |

**Table 1.** *Cont.*

| Instrument | Ocean Relevant Technical Characteristic | Enabled Science Strengths/Challenges |
|---|---|---|
| HSRL Pathfinder | 532 nm cross-polarized channel<br><br>532 nm co-polarized channel<br><br>532 nm HSRL channel<br><br>680 nm chlorophyll fluorescence<br><br>Detectors: high sensitivity/<1 m vertical sampling | • 4-dimensional plankton community reconstruction, primary production, diel marine animal migration, phytoplankton health, physical–biological linkages, ocean color applications, and measurement. Coverage comparable to CALIGOLA.<br>• Improved separation of backscatter and attenuation compared to CALIGOLA through HSRL technique.<br>• More detailed characterization of aerosols compared to CALIGOLA for improved ocean color atmospheric corrections.<br>• Reduced ocean property retrievals and depth penetration compared to CALIGOLA due to absence of 355 nm laser emission. |
| Blue Lidar | 486 nm cross-polarized channel<br><br>486 nm co-polarized channel<br><br>486 nm HSRL channel<br><br>680 nm chlorophyll fluorescence<br><br>Other specifications to be defined | • Similar ocean retrievals as HSRL Pathfinder but with greater depth penetration and lower daytime sunlight contamination.<br>• Lower cost from mass, power, and size reductions<br>• SmallSat compatible. |

Another technological advance that would enhance signal fidelity of ocean profiling lidars at all water-penetrating wavelengths would be improved photon-counting detectors. Significant technology development has been accomplished on single-photon avalanche diode (SPAD) arrays for automotive lidars and other applications. Properly configured SPAD arrays would enable massively parallel high-speed photon-counting capability to accurately quantify strong near-surface signals at the high end of the dynamic range while still preserving high-fidelity, lower-noise measurements at depth (i.e., the low end of the dynamic range). Such detectors can also achieve higher quantum efficiencies that increase the overall precision of the fundamental measurements and retrieved geophysical parameters. Architectures exist to combine the detection of photons and the binning of profile data on a single chip, significantly simplifying detector electronics and lowering power, mass, and volume. Devices meeting requirements for an ocean lidar mission have yet to be developed, but it is more a straightforward engineering challenge, rather than a technological challenge, to modify existing technology for future ocean lidars. Such devices will eventually permit very fast photon counting (i.e., improved vertical resolution), with the only remaining limiting factor being the laser pulse duration, which is typically in the range of 5–15 ns for space-qualified high-power lasers.

### 4. Filling the Gap

The CZCS and CALIOP were both pathfinder missions. While the former sensor was designed for ocean observations and demonstrated success early in operations, a 10-year data gap nevertheless occurred between the end of the CZCS record and the beginning of follow-on ocean color missions. Fortuitously, a similar gap in global ocean lidar retrievals between CALIOP and CALIGOLA may be, at least partially, avoidable. There are a number of space-based lidars that have been recently launched or are planned for launch over the next year, although (much like CALIOP) ocean research was never

taken into consideration in their instrument designs. The first of these is the Ice, Cloud, and Elevation Satellite (ICESat-2) (https://icesat.gsfc.nasa.gov/icesat/index.php, accessed on 2 September 2023) that was launched in 2018 and carries the Advanced Topographic Laser Altimeter System (ATLAS). ATLAS is a 1064 nm and 532 nm emitting lidar that has already been used for ocean applications [11,81,83,98,99] (Figure 2). ICESat-2 offers global coverage of vertically resolved (sub-meter) retrievals at 532 nm, with ocean retrieval depths of at least 15–20 m in clear water (Table 1). The ICESat-2 orbit provides repeat observations every 91 days. The spatial resolution of ICESat-2 data is less than CALIOP's due to the mission's focus on polar regions [100] and daytime ocean retrievals are challenged by sunlight contamination [11]. At the time of this writing, the ICESat-2 observatory remains healthy and carries enough propellent to maintain its current orbit throughout the 2020s [82] and it is still operating with its primary laser (a backup laser was included in the instrument design). While ocean analyses at the global scale have not yet been conducted using ICESat-2, it may be anticipated that ATLAS data will contribute to filling the lidar data gap until CALIGOLA is operational (Figure 2). In addition, the Chinese Daqi 1 (DQ-1) mission (https://space.skyrocket.de/doc_sdat/daqi-1.htm, accessed on 2 September 2023), launched in April 2022, carries the Aerosol-Cloud High-Spectral-Resolution Lidar (ACHSRL) that includes a dual-polarization backscatter channel. ACHSRL's orbit altitude, telescope size, and vertical resolution are the same as CALIOP, but its total laser power at 532 nm is 2.6 times greater than that of CALIOP (due to higher rep rate (40 Hz) and pulse energy (>130 mJ)). The DQ-1 mission, therefore, also has the potential to assist the global science community in filling the lidar ocean data gap between CALIOP and CALIGOLA (Figure 2). Finally, the EarthCARE mission is scheduled to launch in 2024 (https://earth.esa.int/eogateway/missions/earthcare, accessed on 2 September 2023). EarthCARE is a joint European Space Agency (ESA) and Japan Aerospace Exploration Agency (JAXA) mission that includes the 355 nm emission Atmospheric Lidar (ATLID) instrument. ATLID employs the HSRL approach and includes a depolarization channel, but the utility of the measurements for ocean application is questionable due to its coarse vertical resolution (100 m in the air).

## 5. Outlook and Conclusions

The purpose of this manuscript is to encourage the international science community to begin planning for and creating, through shared resources and collaboration, a sustained global record of satellite lidar ocean observations to improve understanding of ecological and biogeochemical processes and their changes beyond that provided by traditional satellite observations. Importantly, this lidar climate data record is envisioned as a critical but complementary element of a broader observing system that entails continued satellite ocean color and polarimetry measurements [2], an expanding fleet of autonomous in situ profiling platforms, and advanced modeling. This latter modeling element will be essential for seamlessly integrating observations across diverse sensors with differing spatial and temporal resolutions.

Range-resolved observations of the Earth system are of fundamental importance for atmosphere, terrestrial, cryosphere, and ocean science. Indeed, the dawn of the ocean satellite lidar era owes its existence to an instrument (CALIOP) designed for atmospheric applications. CALIOP has provided global assessments of ocean ecosystem standing stocks, such as phytoplankton biomass (Figure 3), that are independent of ocean color measurements, as well as new observations of the ocean inaccessible through passive ocean color [6,7]. The potential for extending this CALIOP record until the launch of CALIGOLA will likewise rely on in-orbit assets designed primarily for non-ocean applications (Figure 2). In turn, the CALIGOLA design is being formulated for interdisciplinary purposes, with emphasis on advancing atmosphere and ocean research but also having important terrestrial and cryosphere applications. The current manuscript has purposefully focused primarily on ocean science enabled by CALIGOLA, deferring detailed descriptions of other disciplinary applications to future publications. It is difficult to imagine that any subsequent

ocean-capable lidar missions will not have equivalent or greater cross-disciplinary value. Accordingly, open and international communication and cooperative planning will be invaluable for cultivating an interdisciplinary philosophy as the foundation for future lidar missions that maximize scientific advances from these investments.

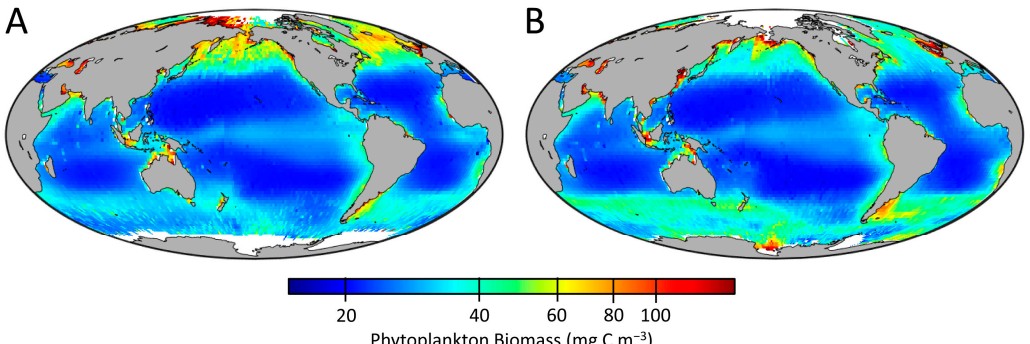

**Figure 3.** Example illustration of seasonal patterns in a key ocean ecosystem property as observed by a satellite lidar. Average (**A**) Boreal summer (June–August) and (**B**) Boreal winter (December–February) surface layer phytoplankton carbon concentrations ($C_{phyto}$) based on CALIOP $b_{bp}$ retrievals at 532 nm ($b_{bp532}$) for the period 2008 to 2017. $C_{phyto}$ was calculated using the field data of Graff et al. [101] but reanalyzed for $b_{bp532}$ rather than the original relationship based on $b_{bp}$ at 470 nm. The resultant new relationship is, $C_{phyto} = 11,510\ b_{bp532} + 3.6$ (n = 45, $R^2$ = 0.70, *p*-value < 0.001). When extended to the maximum of either the euphotic layer depth (i.e., 4.6 optical depths) or mixed layer depth (from http://sites.science.oregonstate.edu/ocean.productivity/, accessed on 15 November 2023), the CALIOP data give an average global phytoplankton standing stock of 0.9 Pg C.

In addition to developing a global satellite lidar climate data record, other lidar missions might be imagined with more targeted objectives. For example, technology advances that reduce instrument size and power requirements may eventually allow for constellations of low-cost SmallSat or even CubeSat [102] lidars characterizing diel cycles in ocean optical properties and reporting on phytoplankton physiology, predator–prey dynamics, and details of animal migrations. Such a lidar constellation might also provide important information on dust deposition processes over the ocean through the provision of detailed and time-resolved measurements of dust and aerosol heights, especially when coupled with atmospheric boundary layer and surface field observations. In this latter case, the benefit of these new measurements may not be global monitoring of deposition but rather a 'process-oriented' assessment where the mass, flux rate, and dust composition are accurately quantified for enough specific events to improve global calculations of deposition in simulation systems. Speculating further, information on surface layer mixing depths might also be a future target for satellite lidar, employing advanced excitation-emission spectroscopy to record temporal changes in rapidly evolving photochemical compounds or new approaches to directly determine depth-dependent changes in water density linked to upper ocean stratification strength.

As we prepare for continued satellite lidar observations of the ocean, it will be important to expand field measurement systems and continue developing next-generation, interdisciplinary science communities prepared to integrate data streams from multiple technologies. To date, approaches to validating lidar ocean products have relied on performance assessments of $b_{bp}$ across a range of spatiotemporal comparisons using Argo observations [17,46,47,99,103]. Looking forward, increasing autonomous sensors from Argo and other platforms will be important to enhance opportunities for validating products from future satellite lidars. In addition, the diversity of these lidar products will expand as satellite technologies advance, so we need to start preparing for a parallel broadening of the suite of in situ validation data, including CDOM absorption, information about particle size distributions, fluorescence quantum yields, microplastic fluorescence, and optical signatures of migrating animals, among other properties described above. Furthermore,

validation of satellite lidar retrievals with Argo data have relied on serendipitous matchups, but looking forward, it may be envisioned how dedicated time series sites that satellite lidars can 'point' at could enable more reliable matchups to improve calibration, validation, and retrieval characterization.

In terms of science community building, open-source software sharing will continue to be crucial to lower access barriers to data processing and enable more effective collaboration across disciplines during the next few years, where an ocean-optimized lidar is not available [104]. There are interdisciplinary studies that can be initiated now not only to advance scientific understanding but to also identify observational requirements that ensure future lidar missions are designed for maximum societal impact. To this end, it will be critical to proactively prepare the next generation of international lidar-literate ocean scientists enabling the success of future missions, including informed advances in lidar technologies, continued science discovery, and societally transformative applications.

The global ocean covers nearly 70% of the Earth's surface and the abundance of organisms living within the surface photic layer alone outnumbers the stars in the sky. These organisms are predominantly planktonic, yet they play a crucial role in global biogeochemical cycles and the provision of food for humanity. In the three-dimensional planktonic world, tight predator–prey coupling results in a rapid tempo of biomass turnover, with local processes having direct consequences on layers above and below that are, in part, mediated by day–night animal migrations. These dynamics cause depth distributions, standing stocks, community composition, and health of marine ecosystems to respond rapidly to environmental change. With the current climate crisis at hand, observing and correctly interpreting these responses are of utmost importance. Here, the unique global perspective provided through satellite remote sensing is crucial but, to date, has been restricted to the surface-most layer of the ocean. The lidar climate data record envisioned herein will enable new dimensions of ocean processes to be revealed that better inform understanding of the Earth system.

**Author Contributions:** Conception: M.J.B. Lidar technical designs: M.J.B., Y.H., C.A.H., P.D.G., D.D., F.L. and S.Z. Manuscript writing/reviewing/editing: M.J.B., L.L., Y.H., K.M.B., C.A.H., P.D.G. and D.D. All authors have read and agreed to the published version of the manuscript.

**Funding:** This work was supported by the National Aeronautics and Space Administration, U.S.A., grant number 80NSSC22K0358.

**Data Availability Statement:** No new data were created for this manuscript except those shown in Figure 3 which are available at http://orca.science.oregonstate.edu/lidar.papers.php.

**Acknowledgments:** The authors thank the reviewers for their constructive and supportive comments, Robert O'Malley for assistance with preparing Figure 3 in the manuscript, and Toby Westberry for assistance in reanalyzing data from Graff et al. [101] to determine the relationship allowing the conversion of CALIOP backscattering data into phytoplankton biomass.

**Conflicts of Interest:** The authors declare no conflict of interest.

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
