# Peer review of "Satellite Lidar Measurements as a Critical New Global Ocean Climate Record"

_remotesensing, doi:10.3390/rs15235567_

Round 1

Reviewer 1 Report

Comments and Suggestions for Authors

This paper promotes the creation of a lidar global ocean climate record and building a global ocean observation network by combining passive remote sensing, in-situ measurement, modeling and other methods. It presents the advantages of spaceborne lidar in ocean observation and the situation of future spaceborne lidars, especially the CALIGOLA mission, and points out that the data gap can be filled by using on-orbit or soon to be launched lidars, such as ICESat-2, DQ-1 and EarthCARE. In my opinion, this paper has a guiding role in building an ocean observation system, developing a new generation of ocean lidar, and promoting ocean data resource sharing and international cooperation.

Author Response

(Comment) This paper promotes the creation of a lidar global ocean climate record and building a global ocean observation network by combining passive remote sensing, in-situ measurement, modeling and other methods. It presents the advantages of spaceborne lidar in ocean observation and the situation of future spaceborne lidars, especially the CALIGOLA mission, and points out that the data gap can be filled by using on-orbit or soon to be launched lidars, such as ICESat-2, DQ-1 and EarthCARE. In my opinion, this paper has a guiding role in building an ocean observation system, developing a new generation of ocean lidar, and promoting ocean data resource sharing and international cooperation.

(Response) We greatly appreciate these encouraging words from Reviewer #1, thank you!

Reviewer 2 Report

Comments and Suggestions for Authors

please see the attached file

Comments on the Quality of English Language

English is fine. Some minor edits are needed and some sentences can be more succinct. 

Author Response

(Comment) This manuscript summarizes the advantages and contributions of satellite Lidar to ocean observation, discusses the development of new satellite lidar missions and their capabilities in ocean observation, and explores the outlook of the role of satellite Lidar data in broader ocean ecological and climate observation. The manuscript is generally well-written and provides some new insight into Lidar remote sensing for oceanography studies. However, there are also some improvements, which have been listed below. Overall, I will recommend a major revision before it is further published.

Major suggestions:

The manuscript focused on the application of satellite Lidar on ocean observation, however, the specific aspect was shifting between the ocean ecosystem (or ecology) and ocean climate. The authors should make it clearer and more straightforward. 

(Response) We have carefully re-examined the manuscript organization and do not agree with Reviewer #2’s conclusion here.  Ocean ecology and ocean climate are intimately intertwined and do not need to be separated when discussing the importance of satellite lidar for Earth system understanding. 

(Comment)  The structure of this manuscript can be improved, the sections in the current version are not well-balanced, and section 4 is too short. The titles of the sections can also be updated. 

(Response) As above, we have carefully re-examined the manuscript and do not concur with this comment by Reviewer #2.  The intent of the Introduction section is to briefly state the overarching theme of the manuscript.  As clearly stated in this Introduction, our intent is not to provide a thorough background on lidar studies of the past, but rather to present a vision of the future in the context of a lidar ocean climate data record.  The second section of the manuscript is focused on explaining to the reader what lidar observation ‘bring to the table’ beyond what is possible with ocean color, as well as complementing ocean color observations.  This is an important part of the manuscript as it provides the scientific justification for promoting international space agencies to cooperate in building a lidar climate data record. Accordingly, this section ‘Lidar Advantage’ is one of the longer sections.  The third section, ‘On the Horizon’ is the heart of the manuscript, describing in detail an envisioned sequence of future lidar missions with increasing observing capabilities than can contribute to construction of a satellite lidar ocean climate data record.  Appropriately, this is the longest section of the manuscript.  Section 4, ‘Filling the Gap’ has a very focused intent: identifying what current and near-term satellite lidar missions can help fill the potential data gap between CALIOP and CALIGOLA. There is a large amount of information already available regarding these gap-filling missions (we have provided citations and links for interested readers) and an exhaustive overview of these missions would not be appropriate for a manuscript focused on future missions beyond those already in space or soon to be in space.

(Comment)  A conclusive section, or some paragraphs, at the end are suggested to summarize the main contributions of these.

(Response) Section 5 entitled ‘Outlook’ already provides 6 concise paragraphs overviewing our conclusions and providing additional forward-looking opportunities for satellite lidar missions.  An additional Conclusion section would seem redundant.

Minor suggestions:

(Comment) Section 2 discussed the advantage of Lidar in ocean observation. However, it is only compared with satellite ocean color. Comparisons with other remote sensing sensors, such as synthetic aperture radar (SAR), and scatterometer, should also be made.

(Response) One of the important aspects of satellite lidar missions is that they have the potential (if appropriately designed) to simultaneously serve many science communities.  We have commented multiple times throughout our manuscript about this interdisciplinary potential, including a call-out box regarding advances in atmospheric science enabled by CALIGOLA.  With respect to Reviewer #2’s comment above, another lidar retrievable geophysical property is ocean surface winds, and lidar-based wind retrievals could certainly be compared to products from scatterometers and/or SAR. Scatterometers normally have a spatial resolution of several km, equivalent to hourly mean winds. SAR can have resolutions as good as 1 km, equivalent to 10 minute averaged wind. Lidar sea surface wind measurements have spatial resolutions of 100 m, equivalent to 10 second averaged wind.  Thus, we agree with Reviewer #2 that there are potential synergies between these technologies.  However, the focus of the current manuscript is on satellite lidar retrievals of in-water ocean geophysical properties and, similar to terrestrial, atmosphere, and cryosphere lidar applications, a discussion on wind retrievals would not be appropriate for this manuscript.

(Comment)  More specific examples should be made to exhibit how satellite Lidar data are used in Oceanography

(Response) This was a great suggestion by the reviewer and is consistent with a comment by Reviewer #3.  Accordingly, we have added a new figure in Section 5 providing an illustrative example of one key ocean ecosystem property retrieved with satellite lidar; specifically, phytoplankton biomass.  We are excited about this addition as it will help readers return to the ‘big picture’ as they get to the end of the manuscript.

Reviewer 3 Report

Comments and Suggestions for Authors

I was quite impressed with this paper.  It makes the case for using lidar measurements for monitoring the effects of climate change and determining e.g., plankton levels in the ocean.  The case was well-made and the paper advocates for international collaborations for the acquisition and analysis of lidar data.  I would have liked to see one or two more figures, e.g., of sample lidar data and how it is interpreted.  But overall, I think the manuscript is worthy of publication and don't think it should be greatly delayed.

Comments on the Quality of English Language

Appears to be mostly fine, but please re-check to make sure no English errors are present.

Author Response

(Comment) I was quite impressed with this paper.  It makes the case for using lidar measurements for monitoring the effects of climate change and determining e.g., plankton levels in the ocean.  The case was well-made and the paper advocates for international collaborations for the acquisition and analysis of lidar data.  I would have liked to see one or two more figures, e.g., of sample lidar data and how it is interpreted.  But overall, I think the manuscript is worthy of publication and don't think it should be greatly delayed.

(Response) We greatly appreciate these encouraging words from Reviewer #3, thank you!  In response to the comment regarding more figures, this was also suggested by Reviewer #2.  Accordingly, we have added a new figure in Section 5 providing an illustrative example of one key ocean ecosystem property retrieved with satellite lidar; specifically, phytoplankton biomass.  We are excited about this addition as it will help readers return to the ‘big picture’ as they get to the end of the manuscript.

Round 2

Reviewer 2 Report

Comments and Suggestions for Authors

The authors have carefully responded to my comments and I'm satisfied with this version. One tiny suggestion is the head of Section 5 can be changed from "Outlook" to "Outlook and Conclusions" for clearance and accuracy.

Author Response

(comment) The authors have carefully responded to my comments and I'm satisfied with this version. One tiny suggestion is the head of Section 5 can be changed from "Outlook" to "Outlook and Conclusions" for clearance and accuracy.

(Response)  This is an excellent suggestion.  The change has been made to the revised manuscript.  Thank you.